# OpenReview forum: "Noisy but Valid: Robust Statistical Evaluation of LLMs with Imperfect Judges"
_ICLR.cc/2026/Conference — ICLR 2026 Poster_

### Official Review · Reviewer_gSSL · 2025-10-19

**Soundness:** 3
**Presentation:** 1
**Contribution:** 3
**Rating:** 4
**Confidence:** 3

**Summary:**

This paper introduces a well-motivated novel approach at verifying LLM judge capabilities through noisy hypothesis testing with minimal required human annotations. They have a large set of LLM judge scores and a small set of human scores and use these two sets to identify the FPR and TPR of the LLM judge scores in relation to human scores. They are then able to perform noisy hypothesis testing using these parameters to adjust the associated threshold with the target prediction to be either accepting or rejecting a model. They provide the first evaluation of the imperfect judge hypothesis testing setting, evaluating how power depends on judge quality, dataset size, and certification levels. They perform empirical validation on three datasets across a range of LLMs and LLM judges to support their theoretical framework.

**Strengths:**

This paper addresses a very important and significant problem: How do we trust the outcomes of LLM judges if the judge itself is imperfect? The authors do a good job of motivating this problem and outlining related methods that attempt to answer this query. The empirical results of the noisy hypothesis testing with oracle parameters indicates that this formulation has potential for high fidelity acceptance or rejection of downstream models conditioned on the knowledge of the TPR and FPR. The empirical results are well motivated and relatively complete, covering many different model, judge, dataset combinations to assure continuity of results. At a general level, this is a novel approach to a timely concern by many in the LLM evaluation research field.

**Weaknesses:**

Throughout the abstract and the introduction, the contributions are not clear. The authors provide limited analysis of results besides the introduction of raw figures and theoretical framing of the evaluation approach. It would be helpful if authors quantified results numerically. In the introduction "We show our procedures outperform conventional ones in terms of type-II error" could be augmented with what that delta of outperformance is or some other clear way to measure progress.

Figure 1 is generally confusing. Fig 1 A-C needs a legend. Does D indicate that there is only a limited part of the plane where noisy hypothesis testing outperforms traditional hypothesis testing? If so, a discussion of this finding/shortcoming is needed. Figure 2 should have a caption.  Further, while the individual plots show differences between varying values of TPR, FPR, nM and nJ, the impact of these changes are not analyzed. It would be helpful/interesting to more systematically evaluate the whole landscape of parameters such that a recommendation could be given to a practitioner-- if you have this FPR/TPR, this many annotations would be needed for this result vs that result such that people could better understand the role of scaling annotation in this setting.

"PPI baselines typically outperform noisy hypothesis testing; however, PPI baselines do not outperform oracle noisy hypothesis testing; this then suggests there may be scope to improve these baselines via judge modelling." If I understand correctly, this supports that PPI outperforms the recommended noisy hypothesis testing method when the parameters are estimated from human annotation subsets. More discussion is needed about why PPI outperforms HT here. This solidifies that the contribution is the noisy hypothesis testing formulation and not the actual method of evaluation since PPI tends to have less error.

Generally, the presentation of the results make it hard to ascertain any individual trends/takeaways. You could aggregate data and show in a table such that it is easier to clearly tell what the best methods are for each error type and under what settings.

**Questions:**

A baseline of using the human annotations directly to accept or reject would be helpful to see if the only benefit of the judge is increased nJ for statistical significance. If we are using the subset of human annotations to calibrate judge quality, what happens if we just use these subset annotations directly to accept/reject the model?

How does the thresholds change if FPRs are really big (you show a max of FPR=0.25, but don't show a full range of parameter effects)?

Need to discuss why you use hypothesis testing as opposed to other traditional forms of LLM judge evaluation approaches (Cohens kappa, ICC, etc.), or at least how your work is positioned among other traditional LLM judge by human judge calibration methods.

In empirical studies, how does the subset TPR and FPR change? You report the oracle parameters but what about the estimators for TPR and FPR? How much closer to the true TPR and FPR are the subset TPR and FPR for 100 nM than 25 nM? What is the scaling law of nM? Again, the current reporting of information in multiple line graphs makes trends/scaling laws hard to extract systematically from the work.

---

> ### Author Response · Authors · 2025-11-23
> **Response to Reviewer gSSL (Part 1/4)**
>
> We thank Reviewer gSSL for their detailed and thoughtful review. We are particularly grateful for the recognition of soundness and contribution and for confirming that our approach is "novel" and addresses a "very important and significant problem". We have taken the presentation critique very seriously and have made significant revisions to address every point below.
>
> > **W1: Throughout the abstract and the introduction, the contributions are not clear. The authors provide limited analysis of results besides the introduction of raw figures and theoretical framing of the evaluation approach. It would be helpful if authors quantified results numerically. In the introduction "We show our procedures outperform conventional ones in terms of type-II error" could be augmented with what that delta of outperformance is or some other clear way to measure progress.**
>
> We thank the reviewer for this suggestion. We agree that quantification is essential. However, we respectfully reasoned that **line plots remain the most effective medium** to visualize the continuous trends of Type-II error probabilities and, crucially, to identify the specific "crossover regions" where our method outperforms baselines (which depend non-linearly on $\alpha$ and judge quality). We had tried prior submission to report the various results using tables rather than figures, but it became complex to report and showcase the various trends.
>
> To address the reviewer's valid concern for clarity and precise measurement, we have instead:
> * **Enhanced Visuals:** We have updated **Figures 1 and 2** with clear, detailed legends and descriptive captions to make the trends and comparisons effortless to follow.
> * **Refined Presentation:** We have significantly revised the **Abstract** and **Introduction** to improve readability and better frame the paper's contributions.
> * **Rigorous Validation:** We emphasize that the comparison of our method with baseline methods is rigorously validated by the specific boundary conditions visualized in **Figure 1(D)** and **Figure 11(Right)**, theoretically proven in **Theorem 5.4**, and further supported by the new finite-sample analysis in **Appendix D.7**. These elements provide the precise theoretical and empirical justification for the performance gains observed in the plots.
>
> For a comprehensive summary of all revisions, we kindly refer the reviewer to our **General Response**.
>
> > **W2.1: Figure 1 is generally confusing. Fig 1 A-C needs a legend. Does D indicate that there is only a limited part of the plane where noisy hypothesis testing outperforms traditional hypothesis testing? If so, a discussion of this finding/shortcoming is needed. Figure 2 should have a caption.**
>
> We thank the reviewer for their careful reading and valuable feedback.
>
> * **Regarding Figure Clarifications:** We have updated **Figure 1** with clear legends and **Figure 2** with a complete, descriptive caption in the revised manuscript to ensure they are self-explanatory.
> * **Regarding Figure 1(D) Interpretation:** The reviewer's interpretation is exactly correct. As illustrated in the figure, Noisy HT outperforms Direct HT only when the judge's quality surpasses a specific threshold, corresponding to a defined region within the (TPR, FPR) plane. Rather than a shortcoming, this is one of the **key theoretical findings** of our work. This boundary is precisely defined by **Theorem 5.4 (Eq. 9)**, which establishes the exact conditions on judge quality required for our method to be superior. To clarify, we restate the specific condition from Eq. 9 below:
>
>     $$(\text{TPR} - \text{FPR})^2 > \frac{\alpha^2 \cdot \frac{\text{TPR} \cdot (1 - \text{TPR})}{R_M} + (1-\alpha)^2 \cdot \frac{\text{FPR} \cdot (1 - \text{FPR})}{1-R_M}}{R_M \cdot (1 - R_M)}$$
>
>     From this condition, we infer that a powerful judge ($\text{TPR} \to 1, \text{FPR} \to 0$) always satisfies the inequality, ensuring Noisy HT outperforms Direct HT. The theorem further characterizes the decision boundary in the ($\text{TPR}$, $\text{FPR}$) plane (visualized in **Figure 1(D)** and **Figure 11(Right)**), demonstrating that higher judge noise ($\text{FPR}$) can be compensated by higher sensitivity ($\text{TPR}$). Additionally, the condition implies that stricter certification scenarios—specifically, higher $\alpha$ or lower $R_M$—raise the bar for the judge: Noisy HT requires a higher $\text{TPR}$ or lower $\text{FPR}$ to beat the direct baseline in these regimes. This analysis has been explicitly added to the revised paper in the **"Practical Implication"** following the theorem.

---

> ### Author Response · Authors · 2025-11-23
> **Response to Reviewer gSSL (Part 2/4)**
>
> > **W2.2: Further, while the individual plots show differences between varying values of TPR, FPR, nM and nJ, the impact of these changes are not analyzed. It would be helpful/interesting to more systematically evaluate the whole landscape of parameters such that a recommendation could be given to a practitioner-- if you have this FPR/TPR, this many annotations would be needed for this result vs that result such that people could better understand the role of scaling annotation in this setting.**
>
> We thank the reviewer for this insightful question regarding parameter analysis and the role of scaling. We are happy to clarify the parameter landscape and provide specific recommendations for practitioners.
>
> **1. General Rule (Asymptotic Decision):**
> First, we clarify that in the asymptotic limit (Theorem 5.4)—where we assume $n_J \to \infty$ and $n_M$ is sufficiently large such that $n_{M_1} \approx R_M n_M$ and $n_{M_0} \approx (1-R_M) n_M$—the relative performance of the Noisy Test vs. Direct Test is **not affected by the size of $\mathbf{n_M}$**. Instead, the superiority condition depends **entirely on the intrinsic quality of the judge** ($\text{TPR}-\text{FPR}$).
> * *Justification:* We posit that $\mathbf{n_J \to \infty}$ is a reasonable assumption given the scalability of automated evaluation, which facilitates the acquisition of massive noisy datasets.
> * *Guidance:* Under these standard conditions, practitioners should use **Figure 1(D)** as a "Go/No-Go" decision tool: if the judge's quality falls in the "Green Region," the Noisy Test is statistically superior.
>
> **2. Refined Advice (Finite-Sample & Imbalance):**
> To directly address the reviewer's interest in the "role of scaling annotation," we derived a **new finite-sample condition** that retains the specific calibration counts ($n_{M_1}, n_{M_0}$) without assuming $n_M$ is sufficiently large:
> $$(\text{TPR} - \text{FPR})^2 > \frac{n_M}{R_M(1-R_M)} \left[ \frac{\alpha^2 \text{TPR}(1-\text{TPR})}{n_{M_1}} + \frac{(1-\alpha)^2 \text{FPR}(1-\text{FPR})}{n_{M_0}} \right]$$
>
> This formula provides critical guidance for **imbalanced calibration sets**:
> * *The Risk of Imbalance:* If the small calibration set $\mathcal{D}{\_M}$ happens to be highly imbalanced (e.g., containing very few failing examples, making $n_{M_1}$ extremely small), the variance term $\frac{1}{n_{M_1}}$ increases significantly.
> * *Stricter Judge Requirement:* In such "unlucky" scenarios, the "cost" of estimation increases. Consequently, the judge must possess a **significantly higher quality** (a larger gap between $\text{TPR}$ and $\text{FPR}$) to overcome this increased estimation variance and still beat the Direct HT baseline.
>
> We have updated **Appendix D.7** in the revised manuscript to include this refined analysis.

---

> ### Author Response · Authors · 2025-11-23
> **Response to Reviewer gSSL (Part 3/4)**
>
> > **W3: "PPI baselines typically outperform noisy hypothesis testing; however, PPI baselines do not outperform oracle noisy hypothesis testing; this then suggests there may be scope to improve these baselines via judge modelling." If I understand correctly, this supports that PPI outperforms the recommended noisy hypothesis testing method when the parameters are estimated from human annotation subsets. More discussion is needed about why PPI outperforms HT here. This solidifies that the contribution is the noisy hypothesis testing formulation and not the actual method of evaluation since PPI tends to have less error.**
>
> We thank the reviewer for this crucial question, which accurately identifies the core distinction between our work and PPI.
>
> We agree with the reviewer’s assessment: **PPI (especially PPI++ and Ridge PPI) can yield higher statistical power because it is designed as a variance-minimizing estimator.** However, its limitation lies in being a "black-box" correction technique.
>
> A practitioner would opt for our **Noisy Hypothesis Testing** framework to gain **interpretability, diagnostics, and flexibility**, which PPI does not provide. Our unique advantage is the **explicit modeling of the judge's $\text{TPR}$ and $\text{FPR}$**. This design unlocks three critical capabilities:
>
> * **Diagnostic Power:** Our method allows a user to diagnose *why* a certification test fails (e.g., "Is the model unsafe ($R_M$ high)?" vs. "Is the judge blind ($TPR$ low)?"). PPI provides only a final, debiased number without this granular insight.
> * **Flexibility with Priors:** Our framework can easily integrate prior knowledge (e.g., known bounds on a judge's TPR/FPR) to improve estimation. As shown in our new **Appendix E (Bounded Estimation)**, incorporating these simple bounds allows our method to **surpass PPI** in performance in certain regimes, providing a promising direction for closing the "Oracle Gap." Note that it is not clear how to integrate judge priors onto the PPI framework and variants, implying that it is also not clear how to close the gap between PPI and the oracle setting.
> * **Transferability & Cost-Saving:** A calibrated judge profile ($\text{TPR}, \text{FPR}$) is potentially transferable across similar tasks. This implies the expensive human-labeling step ($\mathcal{D}_M$) may not need to be repeated for every new model iteration, offering significant long-term cost savings compared to PPI's model-specific correction.
>
> Therefore, our method is the choice for practitioners who prioritize **diagnostic capability and long-term transferability** over raw variance minimization alone.
>
> > **W4: Generally, the presentation of the results make it hard to ascertain any individual trends/takeaways. You could aggregate data and show in a table such that it is easier to clearly tell what the best methods are for each error type and under what settings.**
>
> We apologize for the difficulties in ascertaining trends from the original manuscript. We have significantly updated the presentation, including **enhanced figure legends and captions** and **expanded textual quantification** of the results, to make these takeaways clear without losing the continuous trend information provided by the plots. We kindly refer the reviewer to our **General Response** for a summary of these revisions. We hope these updates resolve your concerns, and we are happy to engage in further discussions if any issues remain.
>
> > **Q1: A baseline of using the human annotations directly to accept or reject would be helpful to see if the only benefit of the judge is increased nJ for statistical significance. If we are using the subset of human annotations to calibrate judge quality, what happens if we just use these subset annotations directly to accept/reject the model?**
>
> We thank the reviewer for this question and apologize for the confusion. The baseline suggested—"using the subset of human annotations directly to accept/reject the model"—is **exactly what our paper defines as 'Direct Hypothesis Testing' (Direct HT)**. This method is included in all of our experimental plots as the primary baseline (e.g., the solid blue line in Figure 1). We have updated the **Introduction** and **Certification Setting** sections in the revised manuscript to explicitly define this baseline and the comparison early on.
>
> Regarding the performance comparison between Direct HT and our Noisy HT, our paper provides rigorous theoretical proof in **Theorem 5.4**, which precisely defines the condition—based on judge quality, i.e., $(\text{TPR} - \text{FPR})^2$—under which our 'Noisy HT' method is statistically superior (achieves lower Type-II error) to the 'Direct HT' baseline. These theoretical guarantees are further supported by our empirical experiments.

---

> ### Author Response · Authors · 2025-11-23
> **Response to Reviewer gSSL (Part 4/4)**
>
> > **Q2: How does the thresholds change if FPRs are really big (you show a max of FPR=0.25, but don't show a full range of parameter effects)?**
>
> We thank the reviewer for this suggestion regarding the full parameter range.
>
> The statistical power of our test depends critically on the judge's ability to distinguish good from bad, which is captured by the discriminative gap $(\text{TPR} - \text{FPR})$. As $\text{FPR}$ gets "really big" and approaches $\text{TPR}$, this gap shrinks to zero, and the judge's labels become statistically meaningless.
>
> This behavior is empirically validated in our paper:
> * Comparing **Figure 1 (B)** (FPR=0.05) to **Figure 1 (C)** (FPR=0.25) shows that increasing the FPR drastically increases the Type-II error (lowers power).
> * **Figure 1 (D)** clearly shows that as $\text{FPR}$ increases (moving right on the x-axis), the "Green Region" where Noisy HT outperforms Direct HT shrinks dramatically.
>
> To further validate this conclusion and show the full effect as requested, **we have added new experiments in Appendix H specifically covering high-FPR regimes (where $\text{FPR} > 0.25$, including 0.50 and 0.70).** These new results confirm that as $\text{FPR}$ becomes very large, the power of the Noisy Test degrades exactly as predicted by **Theorem 5.4**.
>
> > **Q3: Need to discuss why you use hypothesis testing as opposed to other traditional forms of LLM judge evaluation approaches (Cohens kappa, ICC, etc.), or at least how your work is positioned among other traditional LLM judge by human judge calibration methods.**
>
> We thank the reviewer for this important question regarding the positioning of our framework. Our work is positioned as a **certified alternative** to traditional methods that rely solely on measuring accuracy or agreement (e.g., F1 score, Cohen's Kappa) for calibration.
>
> * **Traditional Calibration (Judge Evaluation):** Methods like Cohen's Kappa or ICC focus on measuring **inter-rater reliability**—quantifying how well the judge agrees with humans. They output a score (e.g., "Kappa = 0.8") to help practitioners decide if a judge is "good enough." However, they do not directly provide a probabilistic guarantee about the *model's* safety or performance.
> * **Our Framework (Model Certification):** We go a step further. We use the small human-labeled set ($\mathcal{D}_M$) to explicitly estimate the judge's error profile (TPR/FPR). Crucially, these error parameters are then **directly integrated into the hypothesis test's critical threshold ($c'_J$)**. This allows us to transform the judge's noisy signals into a **statistically valid decision** (Accept/Reject) regarding the model's true failure rate ($R_M$), ensuring strict Type-I error control.
>
> We have added a clarifying sentence to the **Appendix I: Extended Related Work** section to draw this distinction explicitly.
>
> > **Q4: In empirical studies, how does the subset TPR and FPR change? You report the oracle parameters but what about the estimators for TPR and FPR? How much closer to the true TPR and FPR are the subset TPR and FPR for 100 nM than 25 nM? What is the scaling law of nM? Again, the current reporting of information in multiple line graphs makes trends/scaling laws hard to extract systematically from the work.**
>
> We thank the reviewer for this insightful question regarding estimator stability and scaling.
>
> To address this, we have conducted further experiments and added a new **Section 6.4 (Diagnostic Analysis)** to the revised manuscript. In this section, we explicitly report the empirical mean and variance of the $\hat{\text{TPR}}$ and $\hat{\text{FPR}}$ estimators across varying calibration sample sizes ($n_M$).
>
> Our results confirm that the accuracy of our estimators follows the standard theoretical scaling law for binomial proportions: the standard deviation (error) scales inversely with the square root of the sample size ($\mathcal{O}(1/\sqrt{n_M})$).
>
> We sincerely thank Reviewer gSSL for their time and constructive feedback. We trust that our detailed responses and the significant revisions implemented—specifically the enhanced quantification of results and improved structural clarity—satisfy all concerns and robustly demonstrate the soundness and novelty of our contribution. We are happy to engage in further discussion if there are any remaining concerns.

---

> ### Author Response · Authors · 2025-11-27
> **Invitation for further discussion**
>
> Dear Reviewer gSSL,
>
> Thank you again for your feedback on our submission. We’ve now submitted our rebuttal and hope it addressed your concerns.
>
> If you have any further questions or would like additional clarification, we’d be happy to help.
>
> Best regards,
>
> Authors

---

> > ### Comment · Reviewer_gSSL · 2025-11-27
> >
> > Thank you to the authors for their timely and detailed responses to my questions. The presentation of the paper has been improved significantly and the flow is much more clear, highlighting your contributions in a straightforward way. I appreciate your clarification regarding the interpretability guarantees of this new method. I have adjusted my score appropriately.

---

> > > ### Author Response · Authors · 2025-11-27
> > >
> > > Dear Reviewer gSSL,
> > >
> > > Thank you sincerely for your kind acknowledgment that the revisions have enhanced the paper’s quality—and for updating your rating accordingly. Thank you again for your thoughtful feedback and constructive suggestions.
> > >
> > > Best,
> > >
> > > The Authors of Submission 19056

---

### Official Review · Reviewer_pCQv · 2025-10-31

**Soundness:** 3
**Presentation:** 3
**Contribution:** 3
**Rating:** 8
**Confidence:** 3

**Summary:**

The paper studies LLM-as-a-Judge evaluation when judges are imperfect. It frames certification as one-sided hypothesis testing on a model’s failure rate $R_M$, but runs the test on a large, noisy, judge-labeled set after calibrating the judge’s TPR/FPR on a small human-labeled set. The method maps the threshold $\alpha$ to a judge-space threshold α′=FPR+(TPR−FPR)α, derives a variance-corrected critical value that accounts for uncertainty in $\widehat{\text{TPR}}, \widehat{\text{FPR}}$, and proves finite-sample type-I control with power characterization. Experiments on synthetic data, toxic-comment classification (Jigsaw/Hate Speech), and a safety (SafeRLHF) generative setup broadly match the theory and compare against direct testing, PPI, and an oracle variant.

**Strengths:**

1. Algorithm 1 includes an explicit critical value with variance terms from judge-parameter estimation; type-I control and type-II bounds are proved (Berry–Esseen based).


2. Experiments cover both classification and generative settings with multiple judges; qualitative alignment with theory increases confidence in the claims.

**Weaknesses:**

1. Critical values use normal/Berry–Esseen approximations. There’s no comparison to Wilson/Clopper-Pearson style bounds when $n_M$​ is small or failures are rare—precisely when certification is most needed.



2. The judge prompts and aggregation choices can materially change TPR/FPR. The paper doesn’t investigate prompt variants, majority-vote vs. single-judge sensitivity, or robustness to minor instruction changes, despite known judge prompt sensitivity.

**Questions:**

1. It’s better to provide advice to help practitioners choose sample sizes or decide when the method beats direct testing.



2. Do you observe positive correlation between model failures and judge errors on the human-labeled set (e.g., both struggle on long or sarcastic inputs)? If so, how does that alter variance terms or recommended $n_M$?


3. If the judge is chosen after peeking at results, how should users adjust α\alphaα (e.g., holdout, Bonferroni/Holm) to avoid inflated type-I error?

---

> ### Author Response · Authors · 2025-11-23
> **Response to Reviewer pCQv (Part 1/3)**
>
> We thank Reviewer pCQv for their expert and positive review. We are particularly grateful that the reviewer recognized the technical rigor of our work, specifically the derivation of the "explicit critical value" with variance terms and the Berry–Esseen based proofs of Type-I control. We are happy to address the reviewer's advanced technical points below.
>
> > **W1: Critical values use normal/Berry–Esseen approximations. There’s no comparison to Wilson/Clopper-Pearson style bounds when is small or failures are rare—precisely when certification is most needed.**
>
> We thank the reviewer for this sharp and excellent point. We agree that in the critical "rare event" regime (e.g., safety certification where the failure rate $R_M$ is very low), the Normal approximation is theoretically less stable.
>
> * **Justification of Current Approach:** We chose the Normal/Berry-Esseen approximation primarily because it allowed for the derivation of a **closed-form, interpretable critical value $c_J^{\prime}$** (Eq. 6), which explicitly shows how variance from all three sources ($\mathcal{D}{\_J}$, $n_{M_1}$, $n_{M_{0}}$) is combined. This interpretability was key to our contribution.
> * **Empirical Robustness:** We emphasize that our method **remains valid even when $n_M$ is small**. Our extensive experiments include scenarios with calibration sets as small as $n_M=25$ (e.g., Figures 3, 4, and 5), and the results consistently show that **Type-I error remains effectively controlled** below the significance level. Theoretically, our theorems explicitly quantify the approximation error terms ($\mathcal{O}(n^{-1/2})$), providing transparency on how the guarantee converges. We firmly believe that acquiring approximately **20-25 high-quality human labels** is a **completely reasonable and realistic assumption** for any serious safety certification effort. Our framework effectively leverages this minimal human signal to ensure statistical validity where purely automated methods fail.
> * **Future Work:** The reviewer is entirely correct that using exact binomial bounds (like Clopper-Pearson or Wilson) is a crucial and non-trivial extension. This would likely result in a procedural test rather than a single closed-form critical value, but would offer more robust Type-I error guarantees for very small sample sizes ($n_M$) or rare failures. We have updated the **Limitations** part in the Conclusion section to explicitly discuss this trade-off and identify it as a key direction for future work. Thank you very much for the comment.
>
> > **W2: The judge prompts and aggregation choices can materially change TPR/FPR. The paper doesn’t investigate prompt variants, majority-vote vs. single-judge sensitivity, or robustness to minor instruction changes, despite known judge prompt sensitivity.**
>
> We thank the reviewer for raising this important point. We agree that prompt and aggregation choices critically affect the resulting TPR/FPR. To address this, we have added a new **Section 6.4** explicitly investigating the impacts of prompting styles and aggregations. As expected, we show that **aggregated prompts (Federated Judges)** lead to the most favorable performance with higher TPR and lower FPR.

---

> ### Author Response · Authors · 2025-11-23
> **Response to Reviewer pCQv (Part 2/3)**
>
> > **Q1: It’s better to provide advice to help practitioners choose sample sizes or decide when the method beats direct testing.**
>
> We thank the reviewer for this insightful request. We acknowledge that the original presentation regarding sample size could be clearer. To provide concrete advice, we have expanded our theoretical analysis in **Appendix D.7** to cover both the general asymptotic regime and the specific finite-sample case.
>
> **1. General Rule (Asymptotic Decision):**
> First, we clarify that in the asymptotic limit (Theorem 5.4)—where we assume $n_J \to \infty$ and $n_M$ is sufficiently large such that $n_{M_1} \approx R_M n_M$ and $n_{M_0} \approx (1-R_M) n_M$—the relative performance of the Noisy Test vs. Direct Test is **not affected by the size of $\mathbf{n_M}$**. Instead, the superiority condition depends **entirely on the intrinsic quality of the judge** ($\text{TPR}-\text{FPR}$).
> * *Justification:* We posit that $n_J \to \infty$ is a reasonable assumption given the scalability of automated evaluation, which facilitates the acquisition of massive noisy datasets. While we previously assumed a sufficiently large $n_M$ to ensure the interpretability of our theorem, the next point generalizes this analysis to the finite-sample setting.
> * *Guidance:* Under these standard conditions, practitioners should use **Figure 1(D)** as a "Go/No-Go" decision tool: if the judge's quality falls in the "Green Region," the Noisy Test is statistically superior.
>
> **2. Refined Advice (Finite-Sample & Imbalance):**
> However, to address cases where $n_M$ is not asymptotically large, we derived a **new finite-sample condition** that retains the specific calibration counts ($n_{M_1}, n_{M_0}$):
> $$(\text{TPR} - \text{FPR})^2 > \frac{n_M}{R_M(1-R_M)} \left[ \frac{\alpha^2 \text{TPR}(1-\text{TPR})}{n_{M_1}} + \frac{(1-\alpha)^2 \text{FPR}(1-\text{FPR})}{n_{M_0}} \right]$$
>
> This formula provides critical guidance for **imbalanced calibration sets**:
> * *The Risk of Imbalance:* If the small calibration set $\mathcal{D}{\_M}$ happens to be highly imbalanced (e.g., containing very few failing examples, making $n_{M_1}$ extremely small), the variance term $\frac{1}{n_{M_1}}$ increases significantly.
> * *Stricter Judge Requirement:* In such "unlucky" scenarios, the "cost" of estimation increases. Consequently, the judge must possess a **significantly higher quality** (a larger gap between $\text{TPR}$ and $\text{FPR}$) to overcome this increased estimation variance and still beat the Direct HT baseline.
>
> We have updated the manuscript to include this refined analysis, helping practitioners gauge whether their specific calibration data composition supports the use of the Noisy Test. We also acknowledge the limitation of Normal approximations when $n_M$ is extremely small and highlight exact binomial methods as a key area for future work.

---

> ### Author Response · Authors · 2025-11-23
> **Response to Reviewer pCQv (Part 3/3)**
>
> > **Q2: Do you observe positive correlation between model failures and judge errors on the human-labeled set (e.g., both struggle on long or sarcastic inputs)? If so, how does that alter variance terms or recommended ?**
>
> We thank the reviewer for this insightful question regarding conditional dependence.
> * **Empirical Observation:** In our experiments, we did observe occasional correlations on specific complex examples (e.g., subtle sarcasm or highly adversarial prompts), although these did not manifest as a systematic violation of independence across the broader datasets.
> * **Mechanism of Absorption:** Crucially, our framework can also potentially **absorb** much of this correlation into the judge parameters. For example, if both the model and the judge struggle with "hard" inputs (where $S_M=1$ but the judge returns $S_J=0$), this directly results in a **lower estimated $\widehat{TPR}$** on the calibration set.
>     * *Consequence:* A lower $\widehat{TPR}$ automatically penalizes the critical threshold $c_J'$ (making it stricter).
>     * *Outcome:* The test becomes more conservative (lowering statistical power) rather than invalid. It correctly "fails to certify" because it recognizes the judge's inability to detect these hard failures.
> * **Theoretical Variance:** The variance calculation in our paper $\text{Var}{\_{Paper}}$ (e.g., Eq. 40) is based on the assumption of statistical independence. We agree, however, that if error modes are heavily clustered (violating i.i.d. assumptions relative to input "difficulty"), $\text{Var}{\_{Paper}}$ could possibly underestimate the true uncertainty $\text{Var}{\_{True}}$ if we believe the correlation between judge and model errors is generally positive. This suggests that **stratified calibration** (grouping by difficulty) is a valuable direction for future work to tighten these bounds.
> * **Qualitative Analysis:** To address this, we have added **Appendix H (Qualitative Analysis of Decision Divergences)**, where we present concrete examples (e.g., sarcasm) illustrating how our framework handles—or fails to handle—these correlated error modes compared to the baseline.
>
> > **Q3: If the judge is chosen after peeking at results, how should users adjust $\alpha$ (e.g., holdout, Bonferroni/Holm) to avoid inflated type-I error?**
>
> This is a critical statistical issue, and the reviewer is exactly right. Our framework assumes the judge $J$ is **pre-specified** before the test. If a user engages in "peeking" (testing multiple judges/prompts and selecting the best one post-hoc), the reported Type-I error rate becomes inflated. To avoid this, we recommend the following standard protocols:
> * **Judge Selection via Validation Set:** We strongly suggest that practitioners use a **separate validation set** (distinct from the calibration set $\mathcal{D}{\_M}$) for iterating on judge prompts or selecting the best judge model.
> * **Private Calibration Set:** The calibration set $\mathcal{D}{\_M}$ must be kept **private (held-out)** and used *only* for the final calibration and hypothesis testing steps once the judge configuration is fixed. This ensures that the statistical validity guarantees (Theorem 4.1) remain intact.
> * **Statistical Correction:** If a separate validation set is not available and multiple judges are tested on $\mathcal{D}_M$, practitioners must apply a correction (e.g., **Bonferroni or Holm**) to the significance level $\zeta$ to account for multiple testing.
>
> We have added a discussion of the "Data Hygiene" requirement to the Limitations in the Conclusion section of the revised manuscript to ensure practitioners are aware of these requirements.
>
> We thank Reviewer pCQv once more for their time and their extremely helpful, constructive review. Their thoughtful and technically precise questions have significantly strengthened the rigor and clarity of our work. We trust that the clarifications provided above—particularly regarding the statistical validity in finite-sample regimes and the practical guidance for application—satisfy all concerns and demonstrate the soundness and importance of our contribution. We are happy to engage in further discussion if there are any remaining concerns.

---

> ### Author Response · Authors · 2025-11-25
>
> Dear Reviewer pCQv,
>
> We sincerely appreciate your recognition that the revisions have improved the paper's quality. Thank you again for your thoughtful feedback and constructive suggestions.
>
> Best,
>
> Authors of Submission 19056.

---

### Official Review · Reviewer_A3bd · 2025-11-01

**Soundness:** 3
**Presentation:** 3
**Contribution:** 3
**Rating:** 8
**Confidence:** 3

**Summary:**

Establishing the reliability of an LLM is a challenge to tackle due to factors such as data leakage in benchmarking and human evaluation being costly. Recently the LLM-as-a-Judge paradigm has emerged as a tool to look at the reliability of a model, however this is heavily dependant on the quality of the judge.

This paper develops a hypothesis-testing framework to verify whether an LLM is reliable or not, keeping the noisiness of such a judge in mind. The FPR is characterized as the probability of the LLM-as-a-judge perceiving a correct model response as incorrect and the TPR as the probability that the LLM-as-a-judge perceives an incorrect model response as incorrect.
The type I error probability is characterized as the risk of the LLM where an unreliable LLM is incorrectly seen as reliable and type II error as when a reliable model is incorrectly seen as unreliable.

The null hypothesis $H_0$ is defined as the true failure rate of the model being larger than the user specified threshold $\alpha$.
As we are dealing with a noisy model setup with a noisy LLM-as-a-judge for evaluation, the formulation of the procedure is applied to the noisy LM failure rate, where $\alpha$’ is used instead: $\alpha$’ = FPR + (TPR-FPR)$\alpha$
where FPR and TPR are from the LLM-as-a-Judge.

The procedure relies on two data sources ($D_M$: human verified ground truth, small dataset; $D_J$: LLM-as-a-judge labels, large dataset) and primarily consists of two steps:
1. The LLM-as-a-Judge is first employed on $D_M$ to get the TPR and FPR, which in turn is used to get the adjusted threshold $\alpha$’
2. The hypothesis testing is then done on D_J, to determine whether the null hypothesis can be rejected or not

The paper also contains a theoretical analysis with theoretical guarantees for the procedure in terms of Type I and Type II error probabilities in the scenario where we have access to the true TPR and FPR of the LLM-as-a-judge (oracle) and when we have to estimate it ourselves (the real noisy scenario).
When empirically applying the hypothesis testing to a synthetic setting and classification and generation settings, the theoretical guarantees largely hold. Noisy hypothesis testing outperforms direct hypothesis testing, but does not always outperform other baselines such as PPI. In contrast, the oracle scenario does outperform all.

The framework can help in model comparison and judge selection, which is crucial for LLM reliability.

**Strengths:**

1. This paper tackles a highly relevant problem of determining the reliability of LLMs. Benchmarks do not reveal the true capabilities of an LLM and the bias of an LLM-as-a-Judge also does not provide reliable insights. I thus feel using such a hypothesis testing framework is very useful and the need of the hour where there are so many options for which LLM can be used.
2. The approach is grounded in reality. It uses a mix of small-scale human data (which is expensive to get) and large-scale LLM-as-a-Judge data (which is easier to get); this is highly reflective of most LLM setups. Furthermore, it nicely incorporates the popular LLM-as-a-Judge paradigm to make hypothesis testing more appropriate in such settings.
3. The paper consists of both theoretical guarantees as well as empirical results that tie in nicely both model performance as well as LLM-as-a-Judge capabilities and is done in both a synthetic and real dataset scenario with both classification and generation types.
4. The paper is also nicely structured which helps the reader understand the methodology nicely. Especially Figure 2 is very helpful.

**Weaknesses:**

1. I think discussion around the takeaway of the procedure could be clearer. The findings highlight that noisy hypothesis testing outperforms direct hypothesis testing in certain regimes where the TPR is higher and FPR is lower; what does this mean for the takeaway? An overview of in which scenarios the procedure signs would be very helpful. E.g., also the oracle outperforms all but how realistic is this oracle setting also? It would be nice to incorporate this explicitly.
2. Related to the previous point, but if PPI performs better than noisy hypothesis testing, what are its limitations that would make me opt for noisy hypothesis testing instead?
3. I personally feel that having the Related Work right after the Introduction would be nicer. For me as a reader, seeing PPI used as a baseline was not as straightforward as after reading the Related Work section at the end. It kind of felt like PPI came out of nowhere. A bit more explanation on what PPI is / how it works could also likely help with point 2.
4. Not necessarily a weakness but I think a nice discussion point would be LLM-as-a-judge in the scenario for subjective tasks, which is also an issue with human evaluation; who are we asking. Would this be interesting for future work, would the framework need some adaptation or can it be used as such?

**Questions:**

1. Why do you opt for different LLMs as the base model and different LLMs for LLM-as-a-judge? E.g., Mistral 7B only as LLM-as-a-judge? Is there a specific design choice behind it?
2. Just to ensure if I understood correctly; in Figure 2, $D_M$ is in a separate box but the next steps do not use $D_M$ separately right, only in the mixed $\tilde{D}_M$?
3. There is a typo on line 428; an extra “for” in front of “judges”

---

> ### Author Response · Authors · 2025-11-23
> **Response to Reviewer A3bd (Part 1/3)**
>
> We thank Reviewer A3bd for their positive and insightful review. We are especially grateful for the reviewer's accurate summary of our work and for highlighting its strengths, such as its relevance ("need of the hour"), its practical setup ("grounded in reality"), and its clear structure ("nicely structured," "Especially Figure 2 is very helpful"). We appreciate the opportunity to further clarify the discussion points and questions raised.
>
> > **W1: I think discussion around the takeaway of the procedure could be clearer. The findings highlight that noisy hypothesis testing outperforms direct hypothesis testing in certain regimes where the TPR is higher and FPR is lower; what does this mean for the takeaway? An overview of in which scenarios the procedure signs would be very helpful. E.g., also the oracle outperforms all but how realistic is this oracle setting also? It would be nice to incorporate this explicitly.**
>
> Thanks for this excellent and constructive feedback. We agree that we must be more explicit about the key takeaways for a practitioner. Specifically, the reviewer's questions touch on the two key insights of our paper:
>
> * **When does our procedure "shine"? (The Practical Takeaway)**
>     The reviewer is exactly correct that our method "shines" in specific regimes. Our paper provides a precise answer to when this occurs, both theoretically and visually:
>     * **Theoretically:** Theorem 5.4 (Eq. 9) provides the exact mathematical condition for when our method "shines" (i.e., outperforms Direct HT). In short, our method is successful when the judge's quality, defined by its $(\text{TPR-FPR}){^2}$, is high enough to overcome the statistical noise (variance) introduced by estimating these parameters on the small set $\mathcal{D}_M$.
>     * **Visually:** Figure 1(D) and Figure 11(Right) are the very "overview of scenarios" the reviewer requests. The green region explicitly plots the (TPR, FPR) combinations where our method "shines" and outperforms the baseline.
>     * **Action:** As the reviewer suggests, we have added a "Practical Implication" block after each theorem to explicitly frame these results as the "takeaway" for a practitioner.
>
> * **What is the purpose of the "Oracle" setting? (The Theoretical Takeaway)**
>     The reviewer's intuition is correct: it is not intended to be a "realistic" scenario. Its purpose is to serve as an essential theoretical upper bound on performance. It represents the best one could ever do with a noisy judge, assuming one perfectly knew its TPR and FPR. The main "takeaway" here is the gap: Our experiments consistently show a "considerable performance gap" between this ideal 'Oracle' setting and all practical methods, including ours and PPI. This is a key finding, as it suggests the main bottleneck for all such methods is the variance introduced by judge parameter estimation - please refer to our next point for further insights on this estimation challenge.
>     * **Addressing Realism:** Crucially, we have added new experiments to address the "realism" of this setting. In **Appendix E (Bounded Estimation)**, we introduce a scenario where practitioners have partial prior knowledge (e.g., knowing $\text{TPR} > 0.8$). We show that applying these simple bounded estimates significantly reduces estimation variance and narrows the Oracle Gap. This demonstrates that while the pure Oracle is idealized, it inspires approaches that have the potential to bring practical performance come much closer to it by leveraging reasonable priors.

---

> ### Author Response · Authors · 2025-11-23
> **Response to Reviewer A3bd (Part 2/3)**
>
> > **W2: Related to the previous point, but if PPI performs better than noisy hypothesis testing, what are its limitations that would make me opt for noisy hypothesis testing instead?**
>
> We thank the reviewer for this crucial question, as it gets to the heart of our paper's core contribution and positioning. The reviewer is correct that PPI can show higher statistical power. PPI's limitation, however, is that it is a black-box estimator. We expect a practitioner would opt for our method to gain **interpretability and diagnostics**. Our framework's unique advantage is its **explicit modeling of the judge's TPR and FPR**. This explicit design unlocks several unique advantages that PPI cannot offer:
>
> * **Diagnostic Power:** Our method allows a user to diagnose *why* a certification test fails (is the LLM bad, or is the Judge bad?). PPI only provides a final, debiased number.
> * **Flexibility with Priors:** Our framework can easily integrate prior knowledge (e.g., known bounds on a judge's TPR/FPR) to improve estimation. As shown in our new **Appendix E (Bounded Estimation)**, incorporating these simple bounds allows our method to **surpass PPI** in performance in certain regimes, providing a promising direction for closing the "Oracle Gap." Note that it is not clear how to integrate judge priors onto the PPI framework and variants, implying that it is also not clear how to close the gap between PPI and the oracle setting.
> * **Transferability & Cost-Saving:** A judge's (TPR, FPR) profile is potentially transferable across similar tasks. This means the expensive human-labeling step ($\mathcal{D}_M$) may not need to be repeated, offering significant long-term cost savings.
>
> Therefore, our method is the ideal choice for practitioners who value diagnostic power, flexibility, and cost-saving transferability over raw statistical power alone.
>
> > **W3: I personally feel that having the Related Work right after the Introduction would be nicer. For me as a reader, seeing PPI used as a baseline was not as straightforward as after reading the Related Work section at the end. It kind of felt like PPI came out of nowhere. A bit more explanation on what PPI is / how it works could also likely help with point 2.**
>
> Thank you for this constructive suggestion. We agree it improves the paper's flow, and accordingly, we have relocated the **Related Work section (now Section 2)** to immediately follow the Introduction in the revised manuscript.
>
> To further address the concern that PPI "came out of nowhere," we have also revised the **Abstract and Introduction** to explicitly introduce PPI early on. We added a dedicated discussion in the Introduction distinguishing our framework's design philosophy (validity and diagnostics) from PPI's (estimation). For a comprehensive summary of all structural and content updates, please kindly refer to our **General Response**.

---

> ### Author Response · Authors · 2025-11-23
> **Response to Reviewer A3bd (Part 3/3)**
>
> > **W4: Not necessarily a weakness but I think a nice discussion point would be LLM-as-a-judge in the scenario for subjective tasks, which is also an issue with human evaluation; who are we asking. Would this be interesting for future work, would the framework need some adaptation or can it be used as such?**
>
> Thanks for this interesting suggestion. We agree that the scenario of using an LLM-as-a-judge for subjective tasks (like safety alignment, quality, or harmfulness) is a fascinating area for future work. The current framework's foundation is flexible enough to adapt, with a careful redefinition of the ground-truth binary label ($S_M$). Specifically, our current framework is restricted to simple pass/fail evaluation ($S_M \in \{0, 1\}$). For subjective scenarios, the framework would need adaptation in the following ways, which also addresses the questions raised about label quality:
>
> * **Handling Continuous or Multiple Labels:** To adopt the framework to handle common subjective metrics like continuous or multiple labels (e.g., Likert scales, pairwise ranks, or aggregation of multiple human opinions), the certification goal would shift. One approach is binarization (e.g., score $\ge 4$ is "pass"), but future work could explore adapting the framework and test statistics themselves to handle continuous outcomes.
> * **Modeling Noise in Human Labels:** The reviewer correctly hints at a major underlying issue: in subjective settings, even the human consensus ($S_M$) is inherently noisy. Our current framework models the LLM-judge as imperfect relative to a perfect $S_M$. A more advanced framework for subjective tasks would need to treat the ground truth itself as a probabilistic latent variable, similar to some approaches dealing with noisy labels in related literature.
>
> We have also incorporated this important discussion briefly into our Conclusion section to highlight the path toward rigorous certification in subjective domains.
>
> > **Q1: Why do you opt for different LLMs as the base model and different LLMs for LLM-as-a-judge? E.g., Mistral 7B only as LLM-as-a-judge? Is there a specific design choice behind it?**
>
> Thank you for the question. We are happy to clarify our practical considerations on model configurations. In our experiments, we prefer the LLM-as-a-Judge (e.g., Mistral 7B or LLaMA-3.3-70B) to be a reliable 'evaluator’ model compared to the model under certification (e.g., Qwen2.5-0.5B). This asymmetric choice simulates the common and pragmatic setup in the LLM industry, and aims to approximate the reliable human evaluator.
>
> However, we wish to emphasize that **our framework does not mandate this asymmetric model**. Our design reflects the flexibility of the LLM-as-a-Judge paradigm, and it is entirely possible to:
> * Use a lightweight LLM as the judge for a larger LLM, or vice-versa, to reflect this flexibility.
> * The validity of our hypothesis testing procedure relies only on the accuracy of the estimated TPR and FPR (judge quality), not on the relative size or capability of the models being compared.
>
> > **Q2: Just to ensure if I understood correctly; in Figure 2, D_M is in a separate box but the next steps do not use D_M separately right, only in the mixed $\tilde{D}_M$?**
>
> We are happy to confirm that your understanding is perfectly correct.
>
> > **Q3: There is a typo on line 428; an extra “for” in front of “judges”**
>
> Thanks for your careful reading. We have proofread the paper again and updated all grammatical errors and typos to our best.
>
> We would like to thank Reviewer A3bd for their time and extremely helpful, constructive review. Their thoughtful questions have led us to further enhance the clarity and positioning of our work. We trust that the clarifications provided above satisfy all concerns and demonstrate the soundness and importance of our contribution. We are happy to engage in further discussion if there are any remaining concerns.

---

### Official Review · Reviewer_cLFu · 2025-11-03

**Soundness:** 2
**Presentation:** 1
**Contribution:** 2
**Rating:** 2
**Confidence:** 3

**Summary:**

The paper proposes using hypothesis testing to estimate the LLM-judge's imperfection by calibrating their false positive and true positive rates using a small human-annotated dataset. This is different from prediction-powered inference, which relies on human annotations to model the judes.

**Strengths:**

- The paper addresses an important challenge in evaluating large models, leveraging statistical frameworks such as hypothesis testing.

- The method is compared against a prediction-powered inference approach, and the paper notes that PPI often outperforms both oracle-noisy upper bounds. These are good findings.

**Weaknesses:**

- I believe that relying on a small dataset to calibrate the automatic judges will depend a lot on the task, model, and the quality of the collected data.

- This paper is very challenging to read. I miss very basic motivations and illustrations of the core ideas of the work. For example, none of the figures make the paper accessible.

- The theoretical insights should have made the paper's contributions relevant. On the contrary, these insights are full of jargon and formulations that are not relevant.

**Questions:**

- Could you please provide examples of when the methods are successful and when they fail?

- I strongly recommend that the paper be revised and rewritten with a clear presentation, reducing jargon and ensuring that the methods and findings are understandable to a wider audience. For example, follow a clear structure (with visualisations): how do you calibrate and why? How do you measure this calibration and compare it to PPI as well as noisy conditions? Give concrete examples. The current version of the paper feels like a very long abstract.

---

> ### Author Response · Authors · 2025-11-23
> **Response to Reviewer cLFu (Part 1/2)**
>
> We thank Reviewer cLFu for their time and feedback. We apologize that the reviewer found the paper "challenging to read". We were encouraged, however, by the contrasting feedback from other reviewers (e.g., A3bd, pCQv), who found the paper "nicely structured" and our theoretical derivation a "key strength". This confirms to us that our core contributions are sound, but we still took on board feedback relating to improvement of the paper's organization and presentation to ensure it is accessible to the widest possible audience.
>
> To this end, we have uploaded a **significantly revised manuscript**. Please kindly refer to our **General Response** for the summarization of improvements. Below, we provide a detailed response to each of the reviewer’s concerns:
>
> > **W1: I believe that relying on a small dataset to calibrate the automatic judges will depend a lot on the task, model, and the quality of the collected data.**
>
> We thank the reviewer for this important question and are happy to clarify. Standard LLM evaluation typically faces a dilemma: human evaluation is precise but costly, while scalable "LLM-as-a-Judge" approaches often rest on the implicit and unverified assumption that the judge is reliable. In practice, researchers often use a small set of human labels for basic validation to establish trust [1].
>
> **We take this standard practice one step further.** Rather than using human validation merely as a sanity check, we incorporate the noise quantified by this validation into a theoretically sound hypothesis testing framework. Our core technical contribution is capturing *how* automatic judge calibration depends on the task, model, and data quality via the small ground-truth dataset. We explicitly account for this dependency as follows:
>
> * **Handling Dependency:** Our critical threshold $c'{\_J}$ (Eq. 6) is not a fixed value; it is a function that **explicitly incorporates the variance** from this small calibration set (via terms dependent on $n_{M_1}$ and $n_{M_0}$).
> * **Guaranteeing Validity:** By doing so, our method guarantees **Type-I error control (Theorem 5.1)** even when the calibration data is small. This is the central "Noisy but Valid" contribution of our paper.
> * **On Data Quality:** Regarding data quality, our framework assumes (as is standard in the community) that a small, high-quality human-annotated dataset ($\mathcal{D}_M$) is available for alignment [1, 2]. Our theoretical guarantees provide a rigorous framework to establish that our certification remains statistically valid even if this high-quality dataset is extremely small.
>
> *[1] Boyeau et al. Autoeval done right: Using synthetic data for model evaluation. ICML 2025.*
>
> *[2] Angelopoulos et al. Prediction-powered inference. Science 2023.*
>
> > **W2: This paper is very challenging to read. I miss very basic motivations and illustrations... none of the figures make the paper accessible.**
>
> We thank the reviewer for the feedback on the presentation. We have taken this very seriously and have revised the manuscript to improve clarity:
>
> * **Figures:** We have updated all figures to increase readability. Specifically, **Figures 1 and 2** now feature clear, detailed legends and descriptive captions to ensure each figure is self-explanatory.
> * **Motivations and Illustrations:** The Abstract, Introduction, and the Certification Setting section have been revised to provide further clarifications and illustrations of the core concepts and motivations underpinning our framework.
> * **Actionable Takeaways:** To make our theoretical results more accessible, we added explicit **"Practical Implication"** blocks after each main theorem. These translate mathematical derivations into plain English takeaways for practitioners.

---

> ### Author Response · Authors · 2025-11-23
> **Response to Reviewer cLFu (Part 2/2)**
>
> > **W3: The theoretical insights should have made the paper's contributions relevant. On the contrary, these insights are full of jargon and formulations that are not relevant.**
>
> We would like to clarify the relevance of these contributions, as they are **essential** to the validity and practical application of our framework:
>
> * **Relevance to Validity:** The formulations (e.g., in **Section 5.1** and **Appendix D.2**) are necessary to establish the statistical soundness of Algorithm 1. Concretely, our analysis is required to derive the **variance-corrected critical threshold $c'_J$**. This threshold is the component that correctly accounts for the uncertainty from judge parameter estimation (on $\mathcal{D}_M$), thereby guaranteeing **Type-I error control (Theorem 5.1)**. This guarantee – which establishes one limits the likelihood on declares a unsafe model to be safe–  is the central "validity" claim of our paper.
> * **Relevance to Practice:** These theoretical insights directly provide the practical guidance the reviewer is seeking. For example, **Theorem 5.4 (Eq. 9)** explicitly characterizes the conditions under which our "Noisy HT" method outperforms the "Direct HT" baseline. This directly informs a practitioner *when* our method is useful.
>
> > **Q1: Could you please provide examples of when the methods are successful and when they fail?**
>
> We thank the reviewer for this question. Our paper provides a precise theoretical and empirical answer to when our method is successful:
>
> * **Theoretical Answer:** **Theorem 5.4 (Eq. 9)** provides the exact mathematical condition for when our method "succeeds" (i.e., has higher statistical power than the Direct HT baseline). In short, our method is successful when the **judge's quality**, defined by $(TPR-FPR)^2$, is high enough to overcome the statistical noise introduced by estimating these parameters on the small set $\mathcal{D}{\_M}$.
> * **Visual Examples:** **Figure 1(D)** (and Figure 11, Right) provides the explicit "examples" the reviewer is asking for. These plots visualize the boundary from Theorem 5.4:
>     * **Successful:** The **green region** shows the judge (TPR, FPR) combinations where our method succeeds and outperforms the baseline.
>     * **Fail:** The **red region** shows the combinations where it fails (performs worse than the baseline), typically when the judge is too noisy (low TPR or high FPR).
>
> > **Q2: I strongly recommend that the paper be revised... follow a clear structure... how do you calibrate and why? How do you measure... and compare...?**
>
> We appreciate this advice. The structure the reviewer suggests—(1) How/why we calibrate, (2) How we measure, (3) How we compare—is indeed the core logical flow of our paper, and we have updated the text to make this flow more explicit:
>
> * **'How do you calibrate and why?'**
>     * *How:* **This is detailed in Section 4.2 (Judge Modelling)** and **Figure 2**. We use the small human-labeled set $\mathcal{D}_M$ to estimate $\widehat{TPR}$ and $\widehat{FPR}$.
>     * *Why:* The motivation is to model the judge's noise, which is the necessary input to compute a **statistically valid critical threshold $c'_J$**.
> * **'How do you measure... and compare...?'**
>     * *Measure:* **Section 4.3 (Judge Based Testing) details** how we perform the test using the large judge-labeled set $\mathcal{D}{\_J}$ and our derived threshold.
>     * *Compare:* **Section 6 (Experiments) is dedicated** to the comparison. We systematically compare "Noisy HT" against "PPI" and "Noisy HT (Oracle)" across three datasets. Additionally, **Theorem 5.4** theoretically quantifies exactly when our method wins.
>
> We believe these revisions, along with updated figures and legends, make the paper's structure clearer and directly address the reviewer's concerns. We thank Reviewer cLFu again for their time and feedback, and we are happy to engage in further discussion if there are any remaining concerns.

---

> ### Author Response · Authors · 2025-11-27
> **Invitation for further discussion**
>
> Dear Reviewer cLFu,
>
> Thank you again for your feedback on our submission. We’ve now submitted our rebuttal and hope it addressed your concerns.
>
> If you have any further questions or would like additional clarification, we’d be happy to help.
>
> Best regards,
>
> Authors

---

### Author Response · Authors · 2025-11-23
**General Response**

We sincerely thank the reviewers for their time and constructive feedback. We are encouraged that the reviewers recognized the **significance** and **novelty** of our approach.

### **Summary of Strengths**
We appreciate the consensus on the rigorous nature of our work:
* **Rigorous Foundation:** Reviewers praised the theoretical depth, specifically the derivation of the "explicit critical value" and the "proved finite-sample type-I control."
* **Novelty & Soundness:** The framework was recognized as "well-motivated" and "grounded in reality," addressing the critical gap in trusting imperfect judges.
* **Completeness & Clarity:** Reviewers also commended the empirical completeness covering various models and found the paper "nicely structured."

### **Key Improvements in the Revised Manuscript**
Inspired by the feedback, we have significantly strengthened the manuscript:

* **Enhanced Presentation:** We updated all figures (especially **Figs. 1 & 2**) with clear legends and descriptive captions. We rewrote the **Abstract** and **Introduction** to explicitly frame our contribution as a "diagnostic tool" and moved **Related Work** to Section 2 to improve the narrative flow.
* **Practical Guidance:** We added **"Practical Implication"** blocks after each theorem to explain the theoretical results in plain English. A new **Appendix F** presents concrete case studies comparing *Noisy HT* vs. *Direct HT*, illustrating exactly when our method succeeds or fails. A new **Appendix D.7** further expands our theoretical comparisons of Noisy HT vs. Direct HT to a finete-sample regime.
* **New Experiments:** We added **Appendix E** on "Bounded Estimation" (demonstrating how to reduce the “Oracle Gap” via priors) and **Section 6.4** analyzing estimator scaling laws and prompt robustness. We also included high-FPR stress tests in **Appendix H** to validate theoretical boundaries.
* **Expanded Discussion & Ethics:** We added discussions on the limitations of Normal approximations in small sample regimes and the extension to subjective evaluation tasks. We also updated the **Ethics Statement** to include a content warning for the qualitative examples and to distinguish statistical validity from moral correctness.

We trust these revisions directly address the reviewers' concerns and significantly strengthen the paper. We have provided detailed point-by-point responses to each reviewer below.

*Note on Reference Numbering: Please note that all Theorem, Equation, and Section numbers cited in this response refer to the **revised manuscript** uploaded with this rebuttal, as the structure has been updated to improve clarity.*

---

### Meta-Review · Area_Chair_8WAF · 2026-01-09

**Summary:**

This paper presents a hypothesis-testing framework to address issues with imperfect judges.

Reviewers had some difficulty understanding the core contributions of the paper, which the author response tried to address in their response, and successfully swayed 2 of the lowest scoring reviewers. The reviewer with the lowest score clearly had some issues with understanding the paper, and with the author response it is clear that the paper’s clarity has improved a lot. Keeping in mind all these considerations, I believe the author response does a compelling job of addressing most of the important reviewer concerns.

**Reviewer Concerns:**

**Addressed**
1. Gssl: Lack of Clear Contributions and Quantification, Figure Clarity Issues, Parameter Landscape Analysis, PPI Comparison Discussion, High FPR Range , Hypothesis testing vs Traditional Methods, Estimator Scaling Analysis
2. Pcqv: judge prompt sensitivity and aggregation choices, correlation between model and judge errors, multiple testing/judge selection issues.
3. A3bd: takeaway clarity, PPI vs. proposed method, related work placement, all questions.
4. Clfu: Calibration dependency on task/model/data quality, paper readability and lack of basic motivations/illustrations, theoretical insights appearing irrelevant with excessive jargon, e​​xamples of success/failure cases.

**Unaddressed**
1. Gssl’s concern that the human annotation baseline is used directly for accept/reject decisions.  Authors clarified this is their "Direct HT" baseline, but the reviewer may have been asking for something different or this may indicate the baseline wasn't clearly labeled in the original paper.
2. While authors provided finite-sample analysis requested by Gssl, they still rely heavily on asymptotic conditions and may not have fully delivered the "practitioner's guide" the reviewer requested.

**Reviewer Scores:**

- gssl: The reviewer acknowledges that the response adequately addresses their concerns and increases their score based on it. I also agree with this assessment.
-pcqv: The reviewer responded saying that they will maintain their high score, and won’t increase it further given they are not sure about their reviews.
- a3bd: The reviewer never responded, even though the authors were quite responsive overall, with substantive revisions and new experiments added to address the main concerns. However, their score was already very high to begin with, and could not presumably be higher.
- clfu: This reviewer gave a very low score yet never responded. While the authors made improvements, the reviewer's core criticism was that "the current version of the paper feels like a very long abstract" and needs substantial rewriting for a "wider audience" with reduced jargon. The contrasting positive reviews from other reviewers suggests this reviewer may have had a difficulty understanding the paper, rather than an objective flaw. The authors also try to say this in their response to the area chair.

---

### Decision · Program_Chairs · 2026-01-26

Accept (Poster)